

# In situ-measured benthic fluxes of dissolved inorganic phosphorus in the Baltic Sea

Astrid Hylén[1,2], Nils Ekeroth[3], Hannah Berk[4,5], Andy W. Dale[6], Mikhail Kononets[2,7], Wytze K. Lenstra[8,9], Aada Palo[5], Anders Tengberg[2,10], Sebastiaan J. van de Velde[11,12], Stefan Sommer[5], Caroline P. Slomp[7,8], Per O.J. Hall[2]

[1]Department of Biology, University of Antwerp, Wilrijk, Belgium
[2]Department of Marine Sciences, University of Gothenburg, Gothenburg, Sweden
[3]Sweco Sverige AB, Gjörwellsgatan 22, 112 60 Stockholm, Sweden
[4]NIRAS Sweden AB, Hantverkargatan 11B, 112 21 Stockholm, Sweden
[5]Ensucon AB, Stortorget 6, 222 23 Lund, Sverige
[6]GEOMAR Helmholtz Centre for Ocean Research Kiel, Wischhofstr. 1-3, 24148 Kiel, Germany
[7]Research consultant, Gothenburg, Sweden.
[8]Department of Microbiology, Radboud University, Nijmegen, The Netherlands
[9]Department of Earth Sciences, Utrecht University, Utrecht, The Netherlands
[10]Aanderaa-Xylem, Sanddalsringen 5b, Bergen, Norway
[11]Department of Marine Science, University of Otago, Dunedin, New Zealand
[12]National Institute of Water and Atmospheric Research, Wellington, New Zealand

*Correspondence to*: Astrid Hylén (astrid.hylen@uantwerpen.be)

**Abstract.** Sedimentary recycling of phosphorus is a key aspect of coastal eutrophication. Here, we present data on benthic fluxes of dissolved inorganic phosphorus (DIP) from the Baltic Sea, an area with a long eutrophication history. The presented dataset contains 499 individual fluxes measured in situ with three types of benthic chamber landers at 59 stations over 20 years, and data cover most of the Baltic Sea sub-basins (Hylén et al., 2025; https://doi.org/10.5281/zenodo.14812160). The dataset further contains information about bottom-water dissolved oxygen ($O_2$) concentrations, sedimentary organic carbon (OC) content and sediment type. The DIP fluxes differ considerably between basins depending on OC loading and the level of $O_2$ depletion and generally increase from the coast to the central basins. Several stations have been visited on multiple occasions, also at times with different $O_2$ concentrations, which enables investigation of the immediate effects of shifting bottom-water $O_2$ concentrations on the benthic DIP release. The Baltic Sea-wide benthic DIP release is estimated to be 391 – 489 kton y$^{-1}$ based on a data integration based on sediment type and $O_2$ conditions during three years with varying extents of hypoxia and anoxia (2004, 2013 and 2018). The dataset reveals a lack of flux measurements in winter months, coastal areas and sandy sediments; these should be targeted in future studies. Overall, the data is of high quality and will be important for marine management and studies on mechanisms in benthic phosphorus cycling.



# 1 Introduction

The Baltic Sea (Fig. 1) is a semi-enclosed sea in northern Europe with a long history of impacts from human activities and has been suggested to act as a "time machine" for the future coastal ocean (Reusch et al., 2018). Most notably, large parts of the

35   central Baltic Sea suffer from eutrophication-induced oxygen ($O_2$) depletion (Carstensen et al., 2014; Conley et al., 2009). While the $O_2$ depletion has been argued to sustain the eutrophication through elevated sedimentary phosphorus (P) recycling under hypoxic ($>0 - 63$ µM $O_2$) and anoxic (0 µM $O_2$) bottom water conditions (Vahtera et al., 2007), the magnitude of sedimentary P release in different parts of the Baltic Sea has not previously been quantified at the system scale. Importantly, the basins of the Baltic Sea differ substantially in environmental conditions as well as in input rates and sources of organic

40   matter, macro and micronutrients (of the latter, especially iron; Kuliński et al., 2022). This spatial variability, combined with a long tradition of environmental monitoring in the area (Reusch et al., 2018), makes the Baltic Sea suitable for assessing factors that affect P recycling in sediments.



**Figure 1 a) Map of the Baltic Sea and its subbasins. We do not, in this study, consider the Kattegat and the Belt Sea (situated to the west of the Arkona Basin) as being part of the Baltic Sea due to their nature as a transition zone located outside the Danish straits, although they are listed as subbasins by HELCOM. b) Extent of hypoxic (yellow) and anoxic (red) areas in 2004, 2013 and 2018.**

The varied environmental conditions in the Baltic Sea are largely related to strong salinity gradients, driven by the freshwater input and the connection to the North Sea via the narrow and shallow (~20 m deep) Danish straits (Snoeijs-Leijonmalm et al., 2017). Over 200 rivers discharge into the Baltic Sea, primarily into the northern and eastern parts. While less saline surface water leaves the Baltic Sea through the Danish Straits, a deeper current of more saline water enters the Baltic Sea from the North Sea. The high freshwater input causes an increase in surface salinity from ~2 in the north to ~10 in the south and increasing salinity with depth (~3 – 16), resulting in strong vertical density gradients in the southern (more saline) basins. The tidal amplitude in the Baltic Sea is low (< 10 cm) and is generally masked by sea-level variations driven by changes in atmospheric pressure (Novotny et al., 2006). Due to the shallow depths of the Danish straits, the water exchange is limited, and the water residence time in the Baltic Sea is about 30-40 years (Meier 2007). The average depth of the Baltic Sea is 57 m, but many basins reach water depths over 200 m (Leppäranta and Myrberg, 2009) and several deep points exist (e.g., Gotland deep 249 m, Landsort deep 459 m, Ulvö deep 293 m). Wave-induced resuspension of sediments in the Baltic Sea occurs down to water depths of 40-60 m (Danielsson et al., 2007). This resuspension contributes to the shuttling of particles from shallow erosional areas to deep accumulation areas (Jonsson et al., 1990; Leipe et al., 2011; Nilsson et al., 2021). However, local conditions strongly affect sedimentation and the highest sediment accumulation rates are found in small coastal basins (Mitchell et al., 2021).

The Baltic Sea is divided into three main basins, which in turn consist of multiple subbasins (Fig. 1). The Gulf of Bothnia (Bothnian Bay, the Quark, Bothnian Sea) is situated in the northernmost part of the Baltic Sea. The catchment area consists primarily of forests, and the riverine input of terrestrial material is relatively large (Miltner and Emeis, 2001). The Gulf of Bothnia is the least saline and stratified basin in the Baltic Sea, as sills separate it from the southern basins and decrease the inflow of saline deep water (Snoeijs-Leijonmalm et al., 2017). The Gulf of Finland is situated in the easternmost part of the Baltic Sea. This shallow basin is surrounded mainly by forests and non-arable land but also comprises some agricultural and urban land (Miltner and Emeis, 2001). The basin receives a high freshwater input from the Neva River (17% of the total riverine input; HELCOM, 2018b), causing the stratification to be weak in the eastern part and increase toward the open connection with the Baltic Proper (Snoeijs-Leijonmalm et al., 2017). The Baltic Proper is situated in the southern part of the Baltic Sea and is divided into six sub-basins: the Arkona Basin, the Bornholm Basin, the Gulf of Riga, the Northern Gotland Basin, the Western Gotland Basin and the Eastern Gotland Basin. The catchment area of the basin mainly comprises agricultural land, pastures and forests (Miltner and Emeis, 2001). The basin is strongly stratified due to large differences in salinity between the surface and bottom waters. In addition to minor inflows of saline water from the North Sea, so-called major Baltic inflows (MBIs) currently occur about once every 10 years and renew the bottom water (Fischer and Matthäus, 1996).





The drainage area of the Baltic Sea is nearly five times larger than its surface area and is inhabited by ~85 million people (Snoeijs-Leijonmalm et al., 2017). The population growth and intensification of agriculture increased the nutrient input to the Baltic Sea in the early to mid-1900s (Gustafsson et al., 2012; HELCOM, 2018a; Zillén et al., 2008), which caused a threefold

increase in the primary production in the Baltic Proper and Gulf of Finland (Gustafsson et al., 2012; Savchuk et al., 2008; Schneider and Kuss, 2004). Subsequent organic matter degradation has driven a high consumption of $O_2$, leaving 13% and 16% of the Baltic Proper bottom water hypoxic and anoxic, respectively, in 2022 (Fig. 1; Hansson and Viktorsson, 2023). Despite a considerable decrease in the nutrient load from land since the mid-1980s, there has been no concomitant reduction of the water column P pool (Jilbert et al., 2011; Savchuk, 2018). The stability of the pelagic P pool has been attributed to the

long water residence time, intense sedimentary P recycling, and a decreased capacity of the sediment to retain P due to bottom-water $O_2$ depletion (Conley et al., 2009; Gustafsson et al., 2017; Jilbert et al., 2011; Vahtera et al., 2007). Models suggest that the sedimentary recycling of dissolved inorganic phosphorus (DIP) in the entirety of the Baltic Sea releases 229 kton P $yr^{-1}$ to the water column (Gustafsson et al., 2012). While this estimate of the benthic DIP release represents conditions in the early 2010s, it is nine times higher than the current input of total P to the Baltic Sea via rivers (26 kton P in 2020; HELCOM, 2023).

However, an estimate of the sedimentary P recycling on the Baltic Sea system scale based on the measurements of DIP release from the sediment is currently lacking.

Here, we present a dataset of sediment-water fluxes of DIP from 59 sites in the Baltic Sea, measured in situ with benthic chamber landers between 2001 and 2021. Fluxes measured directly in incubations generally give considerably more accurate estimates of the sediment-water solute exchange than fluxes calculated from pore-water profiles, as the low vertical resolution

of data obtained from pore water can obscure strong chemical gradients (Kononets et al., 2021; Nilsson et al., 2019; Sundby et al., 1986; van de Velde et al., 2023). Furthermore, in bioirrigated sediments, pore-water profiles must be combined with site-specific transport coefficients to obtain reliable fluxes. Flux incubations conducted in situ further have the advantage of more closely replicating ambient conditions, as it can be difficult to maintain low $O_2$ conditions in ex situ incubations. We combine the measured DIP fluxes with seafloor substrate data and the areal extent of bottom-water O2 depletion to calculate

the integrated annual sedimentary DIP release in the entirety of the Baltic Sea.

## 2 Materials and methods

### 2.1 Data collection

Data of DIP fluxes measured in situ with benthic chamber landers have been compiled from published data (Ekeroth et al., 2016a; Hylén et al., 2021; Noffke et al., 2016; Sommer et al., 2017; Viktorsson et al., 2012, 2013), amended with previously

unpublished fluxes. All fluxes were re-evaluated according to the procedure described in section 2.2. Measurements have been conducted in 8 of 14 HELCOM-defined sub-basins, with the highest density of sampling sites in the Western and Eastern Gotland Basins and the Gulf of Finland (Fig. 1, Fig. 2). Three types of benthic chamber landers were used to measure the DIP fluxes: the big and small University of Gothenburg landers (UGOT; Kononets et al., 2021) from Sweden, the ALBEX lander





from the Royal Netherlands Institute for Sea Research (NIOZ) in the Netherlands (Witbaard et al., 2000), and two GEOMAR

Biogeochemical Observatories (BIGO; Sommer et al., 2006) from Germany.

**Figure 2 Map of sediment types and stations sampled using benthic chamber landers in the Baltic Sea. "Not available" marks data excluded (Swedish and Finnish archipelago areas) or data not publicly available; "GoB coastal" is represented by station RA2**

**(section 2.3).**



The sampling sites have been classified according to bottom-water $O_2$ concentrations and sediment type (Table 1). Bottom-water $O_2$ concentrations were measured by optodes: Aanderaa model 3830/3835 (accuracy better than 5% or 8 μM, whichever is greater; Bittig et al., 2018; Tengberg et al., 2006) on the UGOT and BIGO landers, and JFE Advantech model RINKO I

(2% accuracy) on the ALBEX lander. The $O_2$ classification is based on concentrations where oxic is > 63 μM $O_2$, hypoxic is > 0 – 63 μM $O_2$, and anoxic is 0 μM $O_2$ (or < 3 μM, the detection limit of Winkler titrations). The sediment type at each station was obtained from the EMODnet Geology Project seabed substrate data (1: 1 000 000 scale, with the smallest cartographic unit of 4 km$^2$), which follows the "Folk 5" classification scheme (Table A1, Fig. 2). Due to the relatively coarse resolution of the EMODnet data, field observations of sediment cores occasionally indicated that the actual sediment type differed from that

inferred from EMODnet. Thus, adjustments to the sediment type classifications were needed for some stations. Due to the lack of grain size distribution data, measured sediment organic carbon (OC) content was used as a proxy for distinguishing between the sediment types "mixed" and "mud–muddy sand", which contain more and less than 5% gravel, respectively (Kaskela et al., 2019). Gravel in sediment generally indicates that the seafloor is subject to sediment erosion, which typically results in lower OC content than in accumulation sediment (such as mud). We thus assume that mud – muddy sand has a higher OC

content than mixed sediment. Indeed, the median OC content (0-2 cm sediment) at stations classified through EMODnet as consisting of mixed sediment is 1.4%, compared to 8.3% OC for mud – muddy sand. A cut-off value of 3 % OC was chosen to differentiate mud-muddy sand sediment from mixed sediment. Thus, all sediments classified as mixed according to EMODnet but with a higher OC content than 3% were reclassified as mud – muddy sand. Conversely, all sediments classified as mud – muddy sand according to EMODnet but with a lower OC content than 3% were reclassified as mixed.


**Table 1 Description of stations. Note that the station names from references 7 and 8 have been modified; all original station names are included in the accompanying dataset (Hylén et al., 2025). Basins: BP = Baltic Proper, GOB = Gulf of Bothnia, GOF = Gulf of Finland. Subbasins: Bor. B = Bornholm Basin, EGB = Eastern Gotland Basin, BB = Bothnian Bay, BS = Bothnian Sea, GOF = Gulf of Finland, NBP = Northern Baltic Proper, WGB = Western Gotland Basin. Oxygen ($O_2$) status: O = oxic, H = hypoxic, A = anoxic.**
**OC = sedimentary organic carbon content (% dry weight), average of top 0-2 cm. [a]influenced by major Baltic inflow. [b]total carbon. [c]0-1 cm sediment depth. [d]OC samples collected and analysed as in van de Velde et al. (2023).**

| Basin | Subbasin | Station | Lat | Long | Depth (m) | O₂ status | OC (%) | Sediment type | Lander | Data source |
|---|---|---|---|---|---|---|---|---|---|---|
| BP | Arkona | BY2 | 54.975 | 14.099 | 47 | O | 5.8 | Mud-muddy sand | ALBEX | 1, 2 |
| BP | Bor. B | BY5 | 55.468 | 15.477 | 87 | H[a] | 5.6 | Mud-muddy sand | ALBEX | 1, 2 |
| BP | Bor. B | HB1 | 55.833 | 15.283 | 48 | O | 0.6 | Mixed | UGOT | 1, 3 |
| BP | Bor. B | HB2 | 55.767 | 15.467 | 52 | O | 1.4 | Mixed | UGOT | 1, 3 |
| BP | Bor. B | HB3 | 55.800 | 14.950 | 41 | O | 0.7 | Mixed | UGOT | 1, 3 |
| BP | EGB | 311 | 57.442 | 20.725 | 65 | H | 1.0 | Mixed | ALBEX | 1, 2 |
| BP | EGB | A | 57.385 | 19.083 | 59 | O | 0.8 | Mixed | UGOT | 4, 5 |
| BP | EGB | B | 57.472 | 19.269 | 75 | O, A | 5.4 | Mud-muddy sand | UGOT | 4 |



| | | | | | | | | | | |
|---|---|---|---|---|---|---|---|---|---|---|
| BP | EGB | BY15 | 57.320 | 20.050 | 237 | H[a] | 11.4 | Mud-muddy sand | ALBEX | 1, 2 |
| BP | EGB | C | 57.467 | 19.434 | 90 | H | 2.9 | Mixed | UGOT | 4 |
| BP | EGB | D | 57.328 | 19.323 | 129 | H, A, H[a], A[a] | 10.1 | Mud-muddy sand | UGOT | 4, 5 |
| BP | EGB | E | 57.125 | 19.508 | 171 | A, H[a], A[a] | 12.3 | Mud-muddy sand | UGOT | 4, 5, 6 |
| BP | EGB | F | 57.287 | 19.801 | 210 | A, H[a], A[a] | 13.2 | Mud-muddy sand | UGOT | 4, 5, 6 |
| BP | EGB | G1 | 57.445 | 20.729 | 65 | O | 2.3 | Mixed | BIGO | 7, 8 |
| BP | EGB | G2 | 57.368 | 20.602 | 80 | H | 1.7 | Mixed | BIGO | 7, 8 |
| BP | EGB | G3 | 57.352 | 20.590 | 96 | H | 7 | Mud-muddy sand | BIGO | 7, 8 |
| BP | EGB | G4 | 57.346 | 20.574 | 111 | H, H[a] | 6.8 | Mud-muddy sand | BIGO | 7, 8 |
| BP | EGB | G5 | 57.312 | 20.552 | 124 | A, H[a] | 8.2 | Mud-muddy sand | BIGO | 7, 8 |
| BP | EGB | G6 | 57.301 | 20.469 | 141 | A, H[a] | 7.9 | Mud-muddy sand | BIGO | 8 |
| BP | EGB | G7 | 57.355 | 20.486 | 151 | A, H[a] | 7.9 | Mud-muddy sand | BIGO | 7, 8 |
| BP | EGB | G8 | 57.351 | 20.476 | 161 | H[a] | 10.1 | Mud-muddy sand | BIGO | 8 |
| BP | EGB | G9 | 57.351 | 20.471 | 175 | A, H[a] | 12.5 | Mud-muddy sand | BIGO | 7, 8 |
| BP | EGB | H | 57.518 | 19.088 | 44 | O | 0.5 | Coarse | UGOT | 4 |
| BP | EGB | J | 57.481 | 18.992 | 31 | O | - | Mixed | UGOT | 4 |
| BP | EGB[1] | LF1 | 57.983 | 21.281 | 67 | H | 2.4 | Mixed | ALBEX | 1, 2 |
| BP | EGB | U | 57.500 | 20.934 | 50 | O | 0.2 | Sand | UGOT | 4 |
| BP | EGB | V | 57.444 | 20.725 | 65 | H | 1.3 | Mixed | UGOT | 4 |
| BP | EGB | Y | 57.317 | 20.549 | 120 | A | 7 | Mud-muddy sand | UGOT | 4 |
| BP | NBP | GF5 | 59.670 | 22.750 | 52 | O | 0.7 | Mixed | UGOT | 9 |
| BP | NBP | GF6 | 59.600 | 22.550 | 52 | O | 0.7 | Mixed | UGOT | 9 |
| BP | NBP | KH104 | 59.336 | 18.769 | 104 | H, A | 6.6[b] | Mud-muddy sand | UGOT | 10 |
| BP | NBP | KH38 | 59.338 | 18.745 | 38 | O | 2.1[b] | Mixed | UGOT | 10 |
| BP | NBP | KH58 | 59.340 | 18.756 | 58 | O | 6.2[b] | Mud-muddy sand | UGOT | 10 |
| BP | NBP[1] | LL19 | 58.881 | 20.311 | 173 | A | 9 | Mud-muddy sand | ALBEX | 1, 2 |
| BP | NBP | NWBP | 58.437 | 18.423 | 149 | A | 6.2[c] | Mud-muddy sand | UGOT | 11 |
| BP | NBP | PV1 | 59.580 | 22.470 | 73 | O, H, A | 3.7 | Mud-muddy sand | UGOT | 9 |
| BP | WGB[1] | T002 | 58.355 | 17.797 | 120 | A | 12.4[d] | Mud-muddy sand | UGOT | 1 |
| BP | WGB[1] | T003 | 58.274 | 17.749 | 120 | A | 11.9[d] | Mud-muddy sand | UGOT | 1 |
| BP | WGB[1] | T005 | 58.070 | 17.821 | 169 | A | 12.3 | Mud-muddy sand | UGOT | 1, 12 |
| BP | WGB[1] | T006 | 57.979 | 17.909 | 196 | A | 10.8[d] | Mud-muddy sand | UGOT | 1 |
| BP | WGB[1] | T008 | 57.829 | 18.085 | 100 | A | 9 | Mud-muddy sand | UGOT | 1, 12 |





| | | | | | | | | | | |
|---|---|---|---|---|---|---|---|---|---|---|
| BP | WGB[1] | T009 | 57.756 | 18.137 | 112 | A | 11.4 | Mud-muddy sand | UGOT | 1, 12 |
| BP | WGB[1] | T010 | 57.932 | 17.967 | 155 | A | 16 | Mud-muddy sand | UGOT | 1, 12 |
| BP | WGB[1] | T011 | 58.533 | 17.790 | 62 | O | 1.5[d] | Mixed | UGOT | 1 |
| BP | WGB[1] | T012 | 58.523 | 17.798 | 71 | H | 5.2 | Mud-muddy sand | UGOT | 1, 12 |
| BP | WGB[1] | W009 | 57.052 | 17.569 | 110 | A | 13.4[d] | Mud-muddy sand | UGOT | 1 |
| BP | WGB[1] | W010 | 57.078 | 17.482 | 94 | A | 9[d] | Mud-muddy sand | UGOT | 1 |
| BP | WGB[1] | W013 | 57.129 | 17.313 | 72 | H | 10.6[d] | Mud-muddy sand | UGOT | 1 |
| BP | WGB[1] | W015 | 57.144 | 17.272 | 65 | O | 1.2[d] | Mixed | UGOT | 1 |
| GOB | BB[1] | RA2 | 65.730 | 22.447 | 11 | O | 4.8 | Mud-muddy sand | UGOT | 1, 13 |
| GOB | BB[1] | GOB1 | 65.191 | 23.398 | 85 | O | 4 | Mud-muddy sand | UGOT | 1, 13 |
| GOB | BB[1] | GOB2 | 64.194 | 21.993 | 111 | O | 4.5 | Mud-muddy sand | UGOT | 1, 13 |
| GOB | BS[1] | GOB3 | 62.118 | 18.553 | 91 | O | 2.5 | Mixed | UGOT | 1, 13 |
| GOF | GOF | GF1 | 59.600 | 23.670 | 72 | A | 5.9 | Mud-muddy sand | UGOT | 9 |
| GOF | GOF | GF2 | 59.530 | 23.250 | 81 | A | 4.4 | Mud-muddy sand | UGOT | 9 |
| GOF | GOF[1] | GOF5 | 59.952 | 25.184 | 65 | H | 8 | Mud-muddy sand | ALBEX | 1, 2 |
| GOF | GOF | KAS | 59.950 | 24.980 | 54 | O | 3.5 | Mud-muddy sand | UGOT | 9 |
| GOF | GOF[1] | LL3A | 60.074 | 26.305 | 60 | H | 8 | Mud-muddy sand | ALBEX | 1, 2 |
| GOF | GOF | XV1 | 60.270 | 27.230 | 37 | O, H | 0.6 | Mixed | UGOT | 9 |

[1]**This study.** [2]**Hermans et al. (2019).** [3]**Nilsson et al. (2019).** [4]**Viktorsson et al. (2013).** [5]**Hylén et al. (2021).** [6]**Hall et al. (2017)** . [7]**Noffke et al. (2016).** [8]**Sommer et al. (2017).** [9]**Viktorsson et al. (2012).**[10]**Ekeroth et al. (2016a).** [11]**Ekeroth et al. (2016b).**[12]**van de Velde et al. (2023).** [13]**Bonaglia et al. (2017).**

## 145   2.2 Sediment-water flux measurements

The benthic chamber landers used to measure the sediment-water fluxes of DIP have been described in detail elsewhere (Table 2). Here, we briefly summarise the sampling procedures. Specifications of the landers are presented in Table 2. Each chamber incubated an area of sediment and a volume of overlying water. Syringes connected to the chamber withdrew water at pre-set times, with an equal volume of ambient bottom water simultaneously entering the chamber. Sensors continuously measured

the $O_2$ concentrations inside and outside the chambers. The length of the incubation was adjusted depending on the reactivity of the sediment, with shorter incubation times in more reactive areas to minimise the changes to the chemical environment inside the chambers. Depending on the lander system, successful incubations were corroborated by, e.g., measurements of turbidity and pressure within the chambers or retrieval of the incubated sediment at the end of the deployment (Kononets et al., 2021; Sommer et al., 2006; Witbaard et al., 2000).

The samples from the UGOT landers were filtered through pre-rinsed 0.45 μm cellulose acetate filters and were stored at 4 – 6°C or frozen until analysis. Samples from the ALBEX lander were filtered through 0.45 μm nylon filters and were acidified with 5 μL suprapur HCl per ml of sample before storage at 4°C until analysis. The samples from the BIGO landers were stored



at 4 – 6°C before determination of DIP. Concentrations of DIP were determined by the ammonium molybdate method using segmented flow colourimetric analysis (Koroleff, 1983) with an analytical precision better than 3%.


**Table 2 Description of the benthic chamber landers. [a]Kononets et al. (2021), [b]Lenstra et al. (2021), [c]Sommer et al. (2006).**

| Lander system | No. of chambers | Chamber area (cm$^2$) | Chamber volume (L) | Sampling syringes per chamber | Volume withdrawn per sample (ml) | Incubation time (h) |
|---|---|---|---|---|---|---|
| UGOT (small and big)[a] | 2 and 4 | 400 | 4 – 13 | 9 | 55-60 | 11 – 36 |
| ALBEX[b] | 3 | 144 | 1.5 – 2.5 | 5 | 30 | 7 – 8 |
| BIGO (I and II)[c] | 2 and 2 | 651 | 9 – 20 | 8 | ~46 | 29 – 62 |

All DIP sediment-water fluxes were calculated from raw data and were quality-controlled using the R-script FLUXER (Hylén and van de Velde, 2025). In brief, the flux $J$ (mmol m$^{-2}$ d$^{-1}$) was calculated as:


$$J = \frac{dC}{dt} \times H \qquad \text{(Eq. 1)}$$

where $dC/dt$ is the change in concentration in the incubated water over time (mmol m$^{-3}$ d$^{-1}$), and $H$ is the height of the incubated water (m).Positive and negative values of $J$ indicate fluxes directed out of and into the sediment, respectively. The value of $dC/dt$ was obtained from the slope of a simple or quadratic linear regression model fitted to data of time versus DIP

concentrations in the chambers. The choice of model was based on the Akaike information criterion score, corrected for small sample sizes (Hurvich and Tsai, 1989). To ensure a good model fit and investigate whether the assumptions for linear regression were fulfilled, the data quality was evaluated with diagnostic graphs. A plot of residuals versus fitted values was used to assess linearity and homoscedasticity, a scale-location plot to assess homoscedasticity, a plot of residuals versus time to assess whether errors were random (i.e., independent of tie or sample order), a normal quantile-quantile (Q-Q) plot to assess

whether errors were normally distributed. An influence plot was further used to identify points with potentially strong influences on the model, such that the addition or removal of the point changes the model fit drastically. The plot displayed studentised deleted residuals, which indicate outliers (points that do not follow the same trend for the dependent variable as the other points), hat values which identify leverage points (extreme values for the independent variable), and Cook's distance, which is a measure of how strongly single observations influence the estimated model parameters. Outliers were removed from

19% of the incubations; these outliers were associated with known sampling issues or were obvious for other parameters as well (e.g., dissolved inorganic carbon, dissolved inorganic nitrogen) and were clearly distinguishable from the rest of the data. Examples of chamber data are shown in Figure A1.



Following verification of the model and data quality, the initial slope of the regression line was multiplied by the chamber water height to obtain the flux. About 30% of regression models gave slopes with $p \geq 0.05$; these slopes were still used to
calculate fluxes after confirming through visual inspection that the resulting fluxes were not outliers (section 3.1). These regression models with $p \geq 0.05$ were retained in the data set to avoid systematically removing low-to-zero fluxes.

**2.3 Calculation of integrated sedimentary DIP release**

The in situ measured DIP fluxes were upscaled to calculate an integrated annual sedimentary DIP release for the Baltic Sea system. Calculations were carried out for three years with different extents of the $O_2$-depleted area: 2004 (small – 45 740 km$^2$
hypoxic, 32 060 km$^2$ anoxic), 2013 (medium – 51 221 km$^2$ hypoxic, 38 920 km$^2$ anoxic) and 2018 (large – 29 408 km$^2$ hypoxic, 62 214 km$^2$ anoxic). The upscaling was based on the areal proportions of different sediment types (Table 1) under oxic, hypoxic and anoxic bottom water within each sub-basin, calculated with the geographic information system (GIS) software QGIS (version 3.22.1, Fig. A2). Coastal regions with archipelago areas, obtained from HELCOM, were excluded from the seafloor substrate layer due to lack of flux measurements (Fig. 2). The seafloor substrate layer was subsequently intersected with
subbasin borders obtained from the Baltic Marine Environment Protection Commission (HELCOM) and the extent of $O_2$-depleted (hypoxic, anoxic) bottom water (Hansson and Viktorsson, 2021) to calculate the extent of each combination of seafloor type and $O_2$ condition within each subbasin.

The integrated sedimentary DIP release was calculated by summing the average DIP load per combination of sediment type, $O_2$ condition and subbasin. Some parts of the HELCOM area do not have a sediment substrate classification in EMODnet; in
the internal load calculations, it was assumed that the relative partitioning of sediment substrates in the unknown areas was the same as in the remainder of the subbasin. Benthic DIP flux data were not available for each combination of sediment type, $O_2$ condition and subbasin. When flux data for a particular combination were missing for one basin, values from similar conditions in other basins were used (Table A2). Sand and coarse-grained sediments underlying $O_2$-depleted waters are scarce in the Baltic Sea, and no integrated loads were calculated from such areas. Integrated loads were also not calculated for the Gulf of
Riga and the Gdansk Bay, as no in situ flux measurements are available from these areas. Fluxes measured at stations under the influence of MBIs were excluded as they represent an extreme case.

Due to the lack of or low number of measurements in certain areas, we deem it misleading to provide uncertainties for the upscaling calculations. As described above, data for specific combinations of subbasin, sediment type and $O_2$ condition do not always exist and have been replaced with measurements from other basins. The generally similar DIP fluxes in areas with the
same sediment types and $O_2$ conditions (Fig. 6a) justify this approach. When data from one sub-basin is used for other sub-basins, an uncertainty can be calculated for the values used to estimate the average sedimentary DIP release. However, in those cases, the uncertainty does not reflect how accurately the actual DIP flux is estimated in that area. There are also unquantified uncertainties in the extent of the different sediment types and $O_2$-depleted areas, which need to be considered when calculating the uncertainty for the integrated DIP fluxes. Hence, the upscaling calculations should be seen as indicative (section 3.2).



## 3 Results and Discussion

### 3.1 DIP fluxes

Between 2001 and 2021, 499 benthic fluxes of DIP were measured in situ with benthic chamber landers at 59 stations in the Baltic Sea during a total of 112 sampling occasions (station visits), corresponding to 155 lander deployments (Hylén et al., 2025). Of the stations, 49 were in the Baltic Proper, six in the Gulf of Finland and four in the Gulf of Bothnia. Most of the sites comprised mud-muddy sand or mixed sediment, with only one station each representing sandy and coarse sediment (Fig. 3a). All three types of bottom-water $O_2$ conditions (anoxic, hypoxic, oxic) were investigated in the Baltic Proper and the Gulf of Finland, whereas the Gulf of Bothnia was fully oxygenated (Fig. 3b). While measurements from most sites only cover one $O_2$ condition, there is data of two $O_2$ conditions from ten stations and of all three $O_2$ conditions from one station. The majority of stations were located between 30 – 100 m depth, with most stations around 50 m depth (Fig. 3d). Only one station was situated shallower than 20 m depth. Samplings were conducted primarily in spring, summer and early autumn; no sampling occurred in January or December (Fig. 3c).

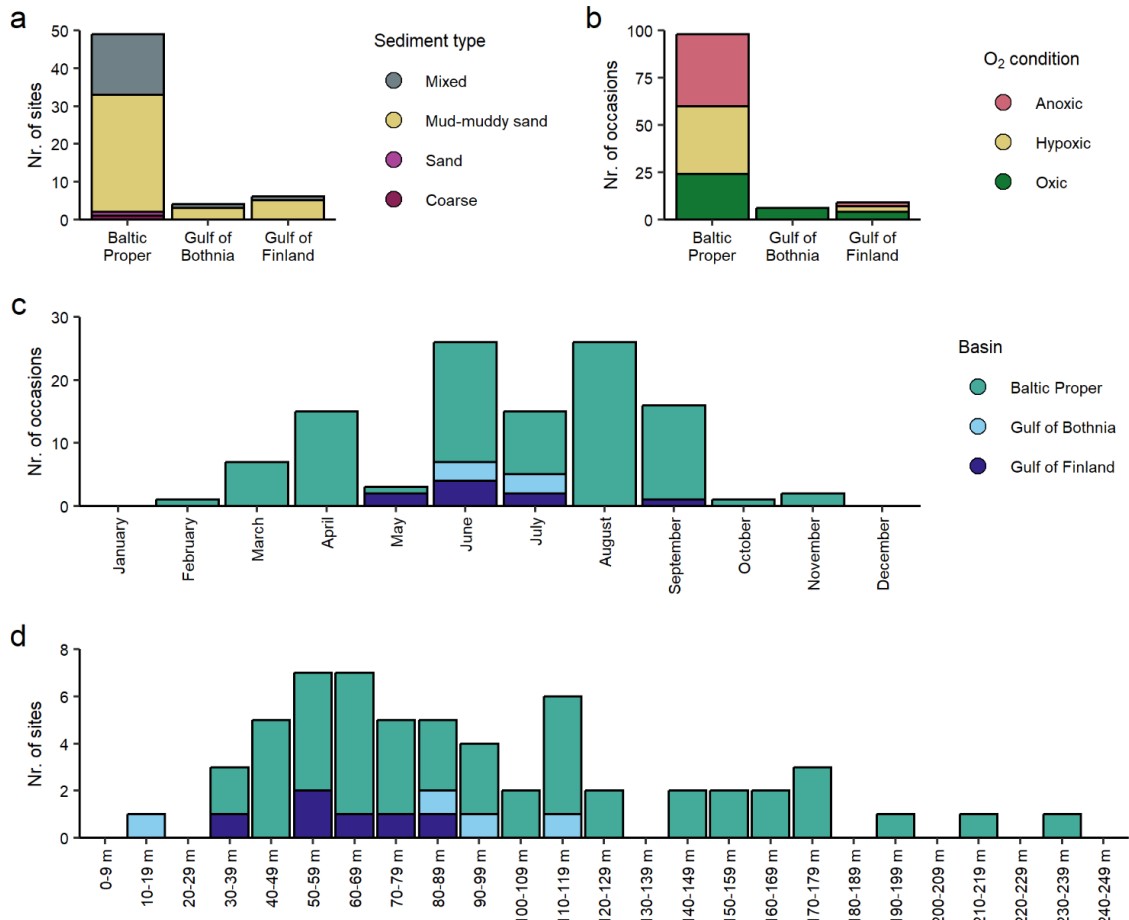

**Figure 3 Distribution of sites and sampling occasions (station visits) over a) sediment types, b) oxygen (O2) condition, c) months, and d) water depth.**


The dataset consists of 499 individual DIP flux measurements (Fig. 4). Of these fluxes, 31% were calculated from linear regressions for which the p-value of the slope was higher than 0.05. Most of these "non-significant" fluxes were close to zero (-0.1 – 0.1 mmol m$^{-2}$ d$^{-1}$) and were measured in shallow, oxygenated areas where the DIP flux is expected to be low, and incubations would have had to be longer to measure a statistically significant change in concentration. Higher "non-significant"

fluxes were in line with (significant) replicates from the same stations and other stations at similar depths and were therefore judged to be accurate. One measurement of good data quality (and p < 0.001) was abnormally high (8.5 mmol m$^{-2}$ d$^{-1}$ at T005, compared to 2.4 mmol m$^{-2}$ d$^{-1}$ in replicate chambers). Strongly elevated fluxes can result from a local deposit of highly reactive organic matter (e.g., a piece of seaweed, a dead fish), which agrees with an unusually high release of dissolved inorganic carbon from the same chamber at station T005 (van de Velde et al., 2023). Although the flux measurements have been

conducted with three different types of landers, data from stations in close proximity sampled by different landers (e.g., the eastern part of the EGB) suggest no systematic deviations depending on the lander type used (Fig. 5).

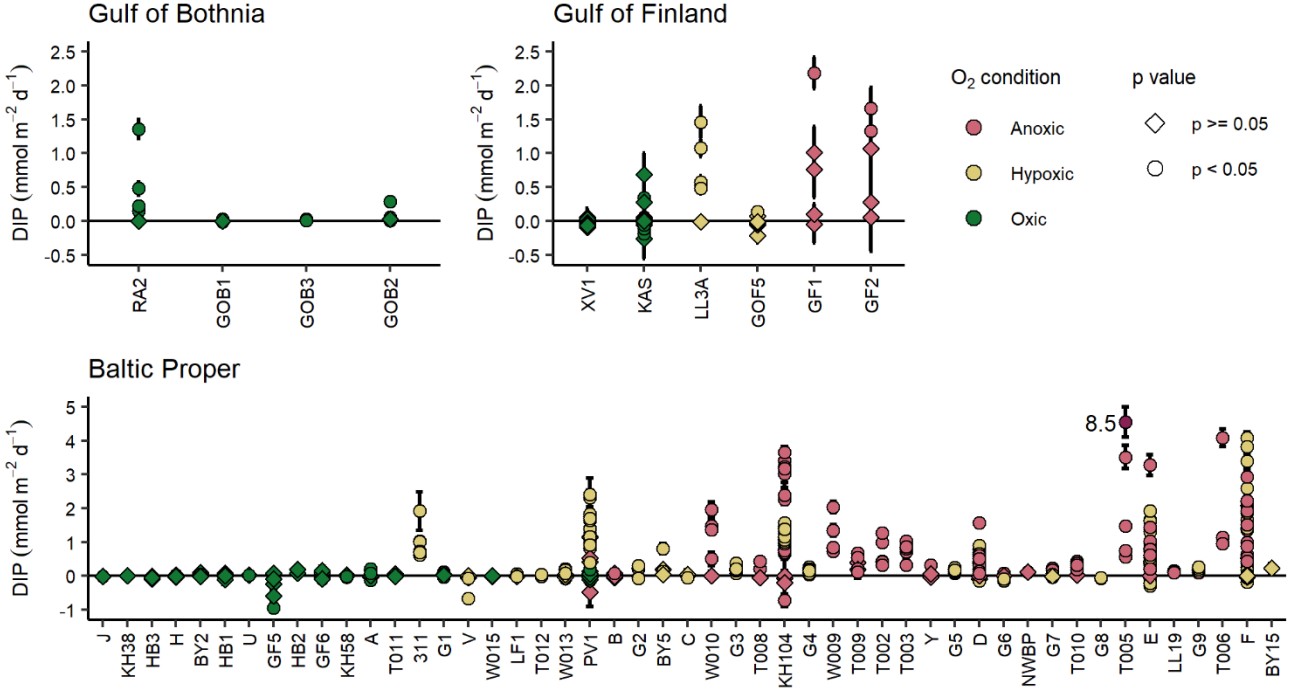

**Figure 4 Individual fluxes of dissolved inorganic phosphorus (DIP) with increasing depth (left to right) and the p-values of the linear**
**regressions used to calculate the fluxes. Each point represents a measurement from one chamber; error bars show the standard error of the flux. A flux of 8.5 mmol m$^{-2}$ d$^{-1}$ measured at station T005 in the Baltic Proper is marked in dark purple.**





**Figure 5 Sediment-water fluxes of DIP in the Baltic Sea, as averages per station. Data from samplings affected by major Baltic inflows are excluded.**



The magnitude of the DIP fluxes differs between subbasins, sediment types and $O_2$ conditions (Fig. 6). The DIP release is generally higher from sediments consisting of mud – muddy sand than from other sediment types, and fluxes are elevated at sites with $O_2$-depleted bottom waters (Fig. 6a). However, stations with mud – muddy sand are overrepresented in the dataset relative to the occurrence of this sediment type in the Baltic Sea (Table A1), and efforts to measure fluxes from other sediment types would likely improve our understanding of spatial differences in the benthic DIP release. Phosphorus dynamics are also strongly impacted by $O_2$ through the adsorption of DIP onto iron oxides, which precipitate in oxic conditions and are reductively dissolved in anoxic conditions (Ruttenberg, 2014; Slomp, 2011). At stations experiencing varying bottom water $O_2$ concentrations, such as stations around the oxycline, it is often unknown how long the site has experienced a specific $O_2$ condition (Fig. 4). In those cases, it is not possible to elucidate whether the measured flux is a short-term response to changing environmental conditions or represents a steady state. Variations in organic matter input can occasionally be detected in the sedimentary DIP release. An increased deposition (and subsequent degradation) of organic matter is seen, for example, in measurements after an MBI in early 2015, which, despite bringing $O_2$ to the central Baltic Proper and initially decreasing the DIP release, later led to an elevated sedimentary DIP efflux due to sediment resuspension and shuttling toward the central part of the EGB (Hylén et al., 2021; Fig. A3).

Despite temporal variations in the sedimentary DIP flux in the Baltic Sea, some spatial trends are clear. The DIP fluxes in the open basins of the Gulf of Bothnia are low and range between 0.01 – 0.06 mmol m$^{-2}$ d$^{-1}$ (Fig. 4-6). The flux is higher (0.44 mmol m$^{-2}$ d$^{-1}$) at a shallow site outside a river mouth in the Bothnian Bay (RA2, 12 m depth). Fluxes in shallow, oxygenated parts of the Baltic Proper and Gulf of Finland are similarly low or even negative, ranging between -0.35 – 0.15 mmol m$^{-2}$ d$^{-1}$. There are two areas of elevated DIP release (1 – 3 mmol m$^{-2}$ d$^{-1}$, Fig. 6b): one at or just below the halocline and the second in the deepest parts of the Baltic Proper basins. This pattern has been attributed to the shuttling of organic and inorganic particles from shallow to deep areas (Jilbert et al., 2011; Nilsson et al., 2019; Turnewitsch and Pohl, 2010). Iron oxides release adsorbed DIP upon reductive dissolution under the hypoxic-anoxic interface prevailing below the halocline, and organic matter is funnelled toward depocenters in the deeper parts of the basin, where it produces DIP when remineralised.



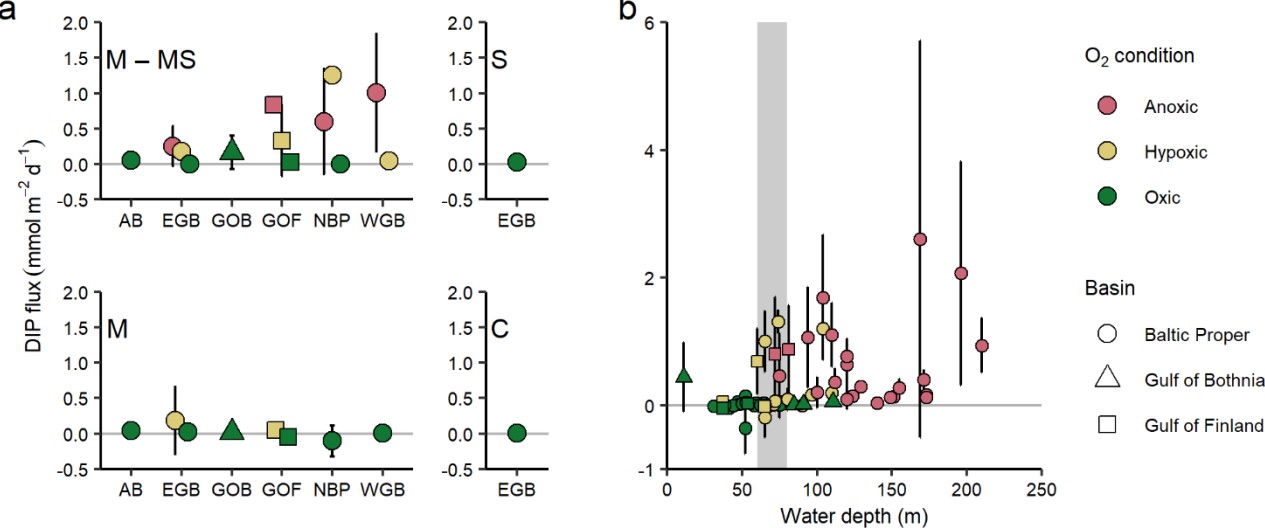


**Figure 6 a) Sedimentary flux of DIP at different sediment types, (sub)basins and oxygen (O₂) conditions. Sediment types: M–MS = mud – muddy sand, M = mixed, S = sand, C = coarse. Subbasins: AB = Arkona and Bornholm Basin, EGB = Eastern Gotland Basin, GOB = Gulf of Bothnia, GOF = Gulf of Finland, NBP = Northern Baltic Proper, WGB = Western Gotland Basin. b) Fluxes of DIP versus water depth. Markers are average values for a station and O₂ condition; error bars show standard deviations. Data from**
**samplings affected by major Baltic inflows are excluded. The grey area marks the approximate depth of the halocline (60 – 80 m).**

### 3.2 Integrated sedimentary release of DIP

The calculated integrated sedimentary DIP release represents 92% of the Baltic Sea area after removal of the Gulf of Riga, the Gdansk Bay, and the archipelago areas in Sweden and Finland (Fig. 2). While the GOB is not impacted by O₂ depletion and

therefore is assumed to have the same DIP flux all three years for which the calculation is carried out, the areal extent of O₂ depletion shifts between years in the other basins. The subbasins in the central Baltic Proper (EGB, NBP, WGB) are most affected by O₂ depletion, and the hypoxic and anoxic areas range from 8 to 30 and 15 to 44% of the total area, respectively (Fig. A3, Table A3). The extent of anoxic and hypoxic areas is reflected in the integrated DIP release, with a higher release when the areal extent of the O₂-depleted water is largest (Fig. 7a). Accordingly, the calculated integrated sedimentary DIP

release for the entirety of the Baltic Sea is 391, 458 and 489 kton y$^{-1}$ during the smallest (2004), medium (2013) and largest (2018) extent of O₂-depleted seafloor, respectively (Fig. 7). Individual subbasins do not always show the same temporal trends in DIP release as the Baltic Sea as a whole, but are dependent on the local extent of O₂ depletion (Fig. A4a). The contribution from the subbasins to the total integrated DIP release further depends on their size (Fig. A4b). While about a quarter of all sedimentary DIP release in the Baltic Sea occurs in the EGB (Fig. 7a), this high contribution is due to the large area of the

EGB since the areal DIP release is considerably lower in this subbasin than in the NBP and WGB (Fig. 7b).





The integrated benthic DIP release calculated here is significantly higher than a previous estimate of the net sedimentary DIP releases based on modelling (229 kton y$^{-1}$; Gustafsson et al., 2012). One reason for this discrepancy may be the few number of data from sites with low DIP fluxes in this study, leading to overestimating the integrated DIP release (section 3.2). However, when comparing these numbers, it should also be noted that the DIP release obtained through our upscaling does not equal the

net amount of DIP becoming available in the water column and surface water since manganese and iron oxide formed at the water column oxycline scavenge a considerable amount of the DIP released from sediments (0.02 – 0.04 mmol m$^{-2}$ d$^{-1}$ in the EGB, or 12 – 25 % of the sedimentary DIP release in hypoxic-anoxic parts of the EGB; Turnewitsch and Pohl (2010)). This mechanism traps the DIP at or below the oxycline and causes intense recycling in the deep parts of the anoxic basins. The calculated integrated sedimentary DIP release in the Baltic Proper is also considerably higher than previous estimates based

on upscaling of in situ measurements (308 – 365 kton y$^{-1}$ vs 110-150 kton y$^{-1}$ from Noffke et al. (2016) and Viktorsson et al. (2013)), but is smaller than a previous estimate from the Gulf of Finland (14 – 33 kton y$^{-1}$ vs 66 kton y$^{-1}$ in Viktorsson et al. (2012)). The upscaling calculations in the previous studies included fewer flux measurements, were based on DIP fluxes averaged over certain depth intervals (without considering the extent of different sediment types), and only accounted for anoxic sediments or depths greater than 60 m in the case of the Baltic Proper.

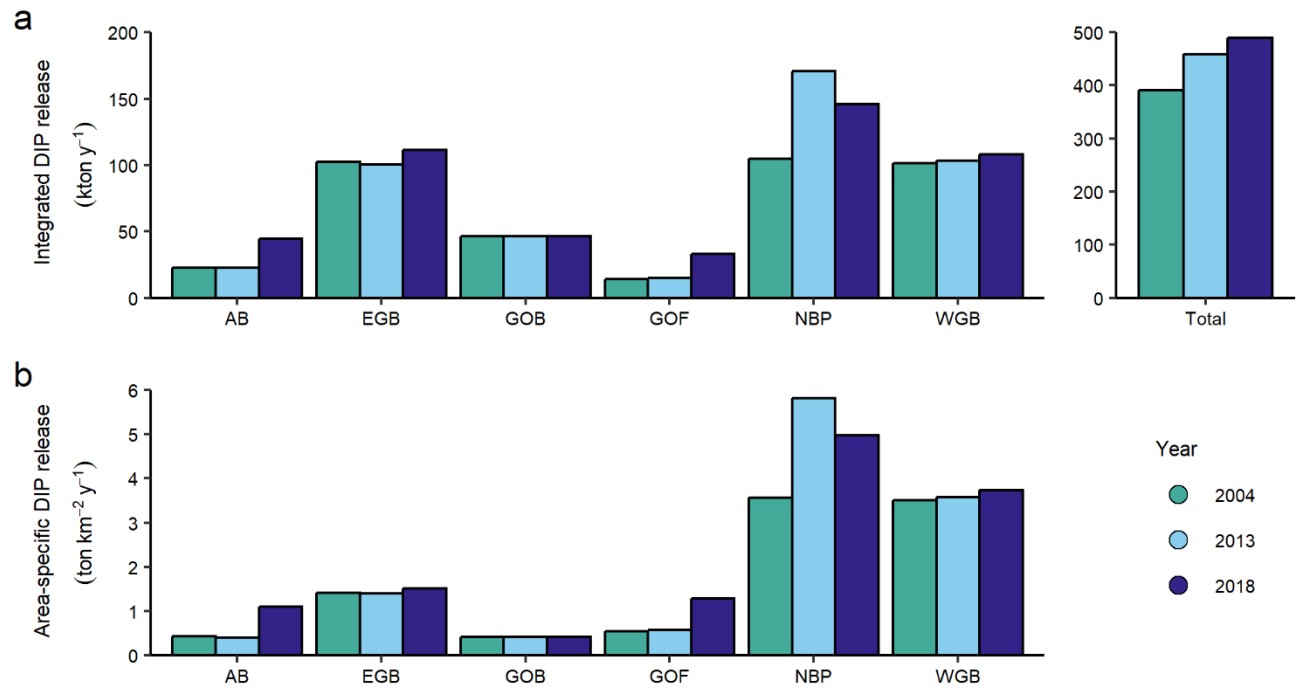


**Figure 7 Basin-wide sedimentary release of dissolved organic phosphorus (DIP) calculated from the in situ measured fluxes for three years with different extents of bottom-water oxygen depletion. Subbasins: AB = Arkona and Bornholm Basin, EGB = Eastern Gotland Basin, GOB = Gulf of Bothnia, GOF = Gulf of Finland, NBP = Northern Baltic Proper, WGB = Western Gotland Basin. a) The integrated sedimentary DIP release. b) Average area-specific sedimentary DIP release.**






## 3.2 Limitations and potential applications

The coverage of in situ DIP flux measurements is good in fine-grained sediments; however, the assembled data reveals a lack of measurements in the Gulf of Riga and the Gdansk Bay and shows that few measurements have been conducted in coastal and shallow areas. Sandy and coarse sediments are also underrepresented in the measurements, likely due to the challenges these substrates pose to in situ benthic chamber incubations (Janssen et al., 2005; Kononets et al., 2021). Furthermore, due to challenging weather and storms which make lander operations risky (Kononets et al., 2021), few flux measurements were conducted during the winter months. The lack of measurements from certain parts of the Baltic Sea and certain seasons might lead to errors in the upscaling calculations. For example, DIP fluxes could be low in coastal sediments where a high input of iron from land, low salinities, and high burial rates cause efficient trapping of P through adsorption to iron oxides and precipitation of vivianite (Carstensen et al., 2020; van Helmond et al., 2020; Slomp et al., 2013). The low primary production and settling of organic matter during the winter months would also expect the benthic DIP flux to be lower than during the rest of the year. Hence, an underrepresentation of potentially low or negative DIP fluxes in the dataset means we might overestimate the integrated sedimentary DIP release. When interpreting the upscaling results, it is also important to note that Baltic Sea sub-basin boundaries according to HELCOM do not reflect natural sediment gradients. Dividing benthic DIP fluxes according to administrative boundaries is relevant from a management perspective but could result in grouping data representing differing environmental conditions.

Since the same sites have not been sampled in more than one to two seasons, the dataset has limited use for interpreting seasonal trends. However, the dataset is sufficiently large to show clear spatial trends in the benthic DIP flux in the Baltic Sea (Fig. 6). The data cover different sediment types, geochemical conditions, salinities, and levels of $O_2$ depletion, making it suitable for validating biogeochemical models. Furthermore, since high water column P concentrations have sustained the eutrophication of the Baltic Sea, data on the input and output of P is critical to deciding on mitigation measures. The presented dataset can be used to locate hot spots of sedimentary DIP release and evaluate the outcomes of mitigation attempts.

## 4 Data availability

All geochemical data described in this paper are available from Zenodo (Hylén et al., 2025; https://doi.org/10.5281/zenodo.14812160). Seabed substrates can be accessed from EMODnet (https://emodnet.ec.europa.eu/geonetwork/srv/eng/catalog.search#/metadata/6eaf4c6bf28815e973b9c60aab5734e3ef9cd9c4) , Swedish and Finnish coastal areas are available from HELCOM (https://metadata.helcom.fi/geonetwork/srv/eng/catalog.search#/metadata/67d653b1-aad1-4af4-920e-0683af3c4a48), and subbasin boundaries from HELCOM (https://metadata.helcom.fi/geonetwork/srv/eng/catalog.search#/metadata/1456f8a5-72a2-4327-8894-31287086ebb5). For the extents of $O_2$-depleted areas, contact SMHI.



## 5 Conclusions

The described dataset combines DIP fluxes measured in situ in the Baltic Sea with key environmental variables regulating benthic phosphorus cycling. The data cover the main subbasins in the Baltic Sea and contain several stations visited on multiple occasions and during varying bottom-water $O_2$ conditions. As such, the dataset gives valuable information about spatial

patterns in the sedimentary DIP release. With this dataset, we present novel estimates of the integrated annual sedimentary DIP release for the entire Baltic Sea, derived from measured fluxes and environmental parameters. The integrated annual sedimentary DIP release was estimated for three different years (2004, 2013 and 2018) representing different areal extents of bottom water oxygen depletion, suggesting a total sedimentary DIP release of $391 – 489$ kton y$^{-1}$. It should be noted, however, that the benthic DIP release does not constitute a new input of P to the Baltic Sea but is the result of internal recycling. The

efflux can only be seen as a net P input to the water column when previously oxic areas become anoxic, and reductive dissolution of iron and manganese oxides releases DIP stored in the sediment. The fluxes measured with the benthic chamber landers are generally of very high quality, and a rigorous statistical quality control procedure has been established to evaluate the data. Measurements conducted by different benthic chamber landers at sites in close proximity further show good agreement. The dataset can thereby reliably be used to investigate mechanisms relevant to benthic phosphorus cycling, to

validate biogeochemical models, and to inform management about areas suitable for eutrophication mitigation attempts.





# Appendix

**Table A1 Sediment type definitions used in the Folk 5 classification scheme (Kaskela et al., 2019).**

| Sediment type (Folk 5) | Definition | Fraction of Baltic Sea seafloor in study (%) |
|---|---|---|
| 1. Mud-muddy sand | Mud 100-10%, Sand <90%, Gravel <5% | 38 |
| 2. Sand | Mud <10%, Sand >90%, Gravel <5% | 15 |
| 3. Coarse sediment | Gravel >80% or Gravel + Sand >95% | 5 |
| 4. Mixed sediment | Mud 95-10%, Sand <90%, Gravel >5% | 38 |
| 5. Rock and boulders | Rock and boulders | 3 |

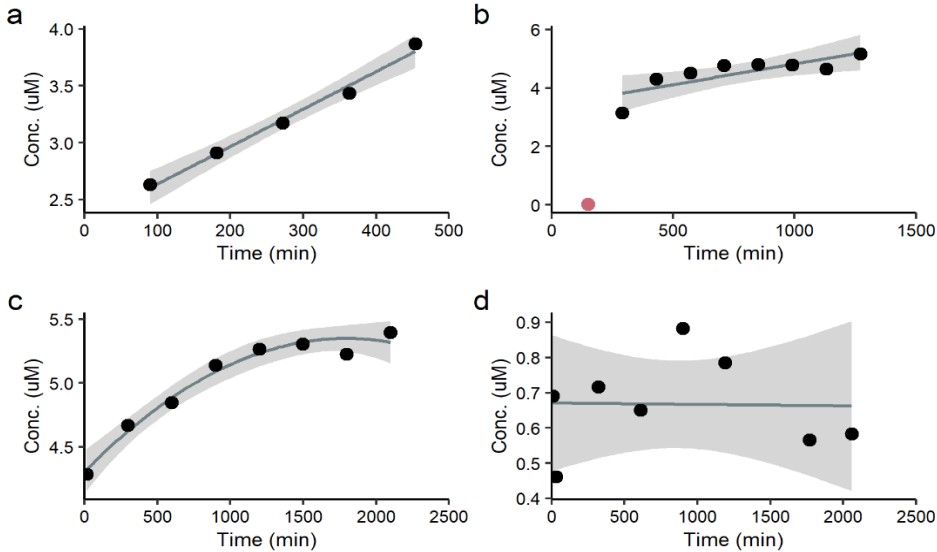

**Figure A1 Examples of data used to calculate fluxes of DIP, showing regression lines (dark grey) with a 95% confidence interval (light grey). a) Station 311 in 2016, ALBEX lander deployment 2, chamber 1. Flux 0.73 mmol m$^{-2}$ d$^{-1}$, simple linear fit, p-value of slope < 0.01. b) Station D in 2008, UGOT Big Lander deployment 1, chamber 3. Flux 0.40 mmol m$^{-2}$ d$^{-1}$, simple linear fit with one outlier (red) removed, p-value of slope < 0.05. c) Station G7 in 2013 BIGO lander deployment 1, chamber 1. Flux 0.23 mmol m$^{-2}$ d$^{-1}$, quadratic fit, p-value of slope < 0.001. d) Station W015 in 2020, UGOT Big Lander deployment 1, chamber 2. Flux -0.001 mmol m$^{-2}$ d$^{-1}$, simple linear fit, p-value of slope 0.96.**





**Figure A2 Schematic description of the QGIS operations to obtain the areas for calculating the integrated sedimentary release of DIP in the Baltic Sea.**



**Table A2 Classification of flux data based on combination of subbasin, oxygen condition and sediment type for calculations of the integrated sedimentary release of DIP in the Baltic Sea.**

| Subbasin | Area | Mud-muddy sand | Sand | Coarse | Mixed | Rock, boulders |
|---|---|---|---|---|---|---|
| GOB | Open waters, oxic | GOB1, GOB2 (23%) | *EGB:* U (7%) | *EGB*: H (2%) | GOB3 (58%) | NA (3%) |
| | Coastal, oxic | RA2 (22%) | *EGB*: U (8%) | *EGB*: H (5%) | GOB3 (46%) | NA (4%) |
| WGB | Oxic | *EGB*: B, *BB*: BY2, *NBP*: KH58, PV1 (7%) | *EGB*: U (1%) | *EGB*: H (1%) | T001 (33%) | NA (8%) |
| | Hypoxic | T012, W013 (5%) | NA (0%) | NA (0%) | *EGB*: C, V, G2; 311, LF1, *GOF*: XV1 (20%) | NA (0%) |
| | Anoxic | NWBP, T002, T003, T005, T006, T009, T010, W009, W010 (23%) | NA (0%) | NA (0%) | | NA (1%) |
| EGB | Oxic | B (14%) | U (14%) | H (9%) | A, J, G1 (18%) | NA (1%) |
| | Hypoxic | G3, G4 (15%) | NA (1%) | NA (1%) | C, V, G2; 311, LF1, *GOF*: XV1 (10%) | NA (0%) |
| | Anoxic | B, D, E, F, Y, G5, G6, G7, G9 (17%) | NA (0%) | NA (0%) | | NA (0%) |
| NBP | Oxic | KH58, PV1 (20%) | NA (0%) | *EGB*: H (1%) | GF5, GF6, KH38 (28%) | NA (6%) |
| | Hypoxic | KH104, PV1 (9%) | NA (0%) | NA (0%) | *EGB*: C, V, G2; 311, LF1, *GOF*: XV1 (12%) | NA (0%) |
| | Anoxic | KH104, PV1, LL19 (24%) | NA (0%) | NA (0%) | | NA (0%) |
| GOF | Oxic | KAS (36%) | *EGB*: U (11%) | *EGB*: H (6%) | XV1 (25%) | NA (6%) |
| | Hypoxic | GOF5, LL3A (7%) | NA (1%) | NA (0%) | XV1; *EGB*: C, V, G2, 311, LF1 (2%) | NA (0%) |
| | Anoxic | GF1, GF2 (4%) | NA (0%) | NA (0%) | | NA (0%) |
| BB | Oxic | BY2, *EGB*: B (13%) | EGB: U (28%) | EGB: H (6%) | HB1, HB2, HB3 (32%) | NA (0%) |
| | Hypoxic | *EGB*: G3, G4 *WGB*: T012, W013 (11%) | NA (0%) | NA (0%) | *EGB*: C, V, G2; 311, LF1, *GOF*: XV1 (3%) | NA (0%) |
| | Anoxic | *EGB*: B, D, E, F, Y, G5, G6, G7, G9; *WGB*: NWBP, T002, T003, T005, T006, T008, T009, T010, W009, W010 (4%) | NA (0%) | NA (0%) | | NA (0%) |


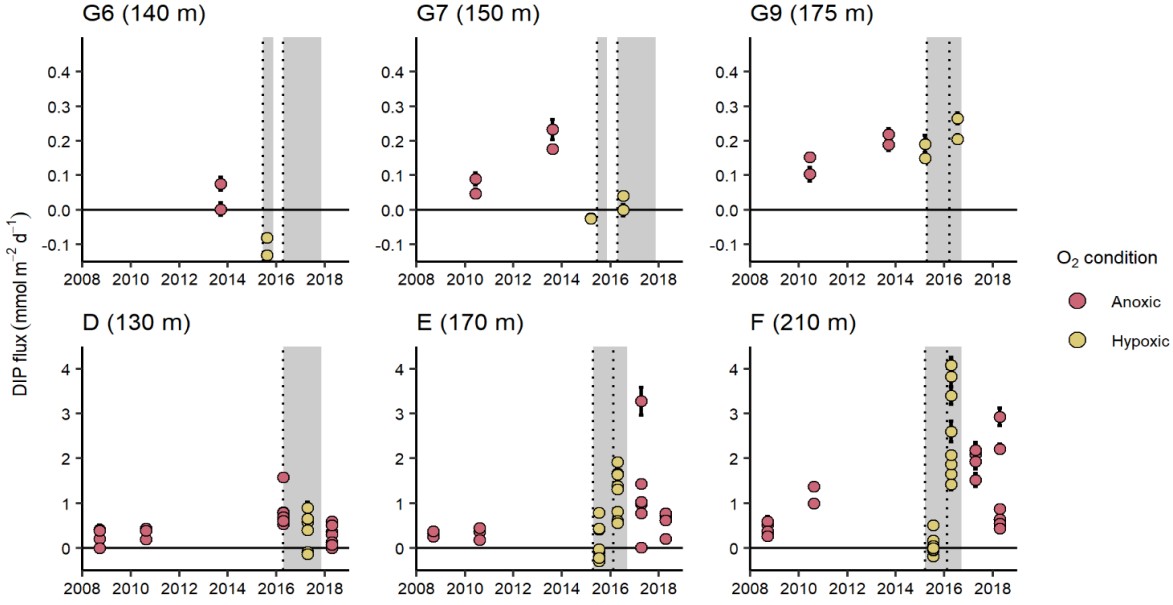

**Figure A3 Fluxes of DIP at stations affected by major Baltic inflows (MBIs). Stations G6, G7 and G9 are situated on the eastern side of the Eastern Gotland Basin, and stations D, E and F are situated on the western side and central part of the Eastern Gotland Basin. The timings of inflows are marked with dotted lines, and periods with oxygen (O2) in the water are marked with grey shading (water column O₂ data from station BY15, SHARKweb SMHI, downloaded 2024-11-20). Error bars show the standard error of the flux.**

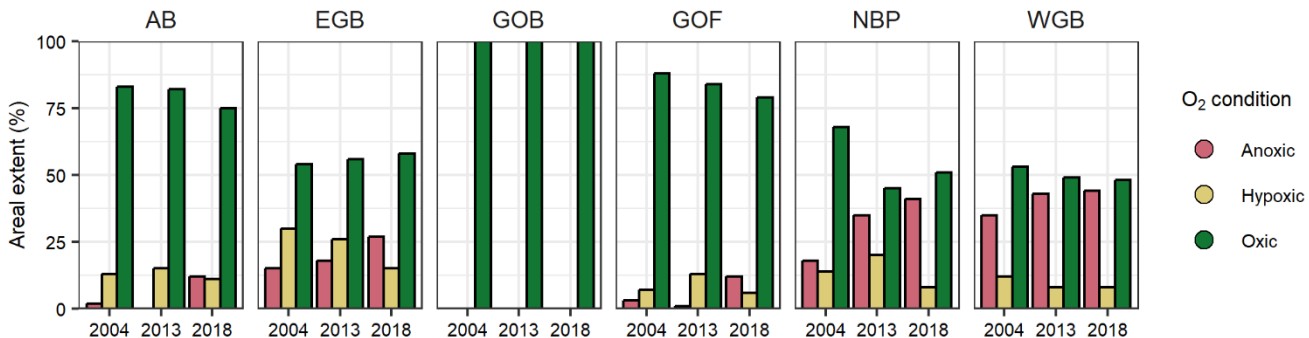

**Figure A4 Percentage of bottom area with anoxic, hypoxic and oxic bottom water in the three years used to calculate the integrated sedimentary release of DIP in the Baltic Sea. Subbasins: AB = Arkona and Bornholm Basin, EGB = Eastern Gotland Basin, GOB = Gulf of Bothnia, GOF = Gulf of Finland, NBP = Northern Baltic Proper, WGB = Western Gotland Basin.**



**Table A3** Areas and resulting integrated fluxes of DIP for individual combinations of sub basin, sediment type and oxygen (O₂) conditions. S = sand, C = coarse, M = mixed, M-MS = mud-muddy sand, O = oxic, H = hypoxic, A = anoxic, HA = hypoxic and anoxic.


| Sub basin | Sed. type | O₂ cond. | Area 2004 (km²) | Area 2013 (km²) | Area 2018 (km²) | Int. flux 2004 (kton y⁻¹) | Int. flux 2013 (kton y⁻¹) | Int. flux 2018 (kton y⁻¹) |
|---|---|---|---|---|---|---|---|---|
| AB | S | O | 19007 | 18999 | 11637 | 5.44 | 5.43 | 3.33 |
| AB | C | O | 4270 | 4269 | 2392 | 0.06 | 0.06 | 0.03 |
| AB | M | O | 16226 | 16069 | 13021 | -1.43 | -1.42 | -1.15 |
| AB | M | HA | 905 | 1062 | 1631 | 2.48 | 2.91 | 4.47 |
| AB | M-MS | O | 6179 | 9077 | 3756 | 2.50 | 3.67 | 1.52 |
| AB | M-MS | H | 4442 | 7774 | 3309 | 6.59 | 11.53 | 4.91 |
| AB | M-MS | A | 970 | 83 | 4432 | 6.84 | 0.59 | 31.23 |
| EGB | S | O | 10409 | 9947 | 11044 | 2.98 | 2.84 | 3.16 |
| EGB | C | O | 6411 | 6396 | 7196 | 0.09 | 0.09 | 0.10 |
| EGB | M | O | 12604 | 15092 | 12590 | 3.88 | 4.64 | 3.87 |
| EGB | M | HA | 8859 | 6371 | 8872 | 24.30 | 17.48 | 24.34 |
| EGB | M-MS | O | 10578 | 9841 | 11808 | 0.32 | 0.29 | 0.35 |
| EGB | M-MS | H | 13236 | 12706 | 6811 | 27.24 | 26.15 | 14.02 |
| EGB | M-MS | A | 10213 | 11480 | 15408 | 43.50 | 48.90 | 65.63 |
| GOB | S | O | 7541 | 7541 | 7541 | 37.52 | 37.52 | 37.52 |
| GOB | C | O | 1804 | 1804 | 1804 | 0.02 | 0.02 | 0.02 |
| GOB | M | O | 62694 | 62694 | 62694 | 0.86 | 0.86 | 0.86 |
| GOB | M-MS | O | 30786 | 30786 | 30786 | 6.05 | 6.05 | 6.05 |
| GOB coast | S | O | 792 | 792 | 792 | 0.23 | 0.23 | 0.23 |
| GOB coast | C | O | 466 | 466 | 466 | 0.13 | 0.13 | 0.13 |
| GOB coast | M | O | 4536 | 4536 | 4536 | 0.89 | 0.89 | 0.89 |
| GOB coast | M-MS | O | 2113 | 2113 | 2113 | 0.77 | 0.77 | 0.77 |
| GOF | S | O | 3288 | 3128 | 3252 | 0.94 | 0.89 | 0.93 |
| GOF | C | O | 1688 | 1639 | 1658 | 0.02 | 0.02 | 0.02 |
| GOF | M | O | 7456 | 7161 | 6566 | -3.99 | -3.83 | -3.51 |
| GOF | M | HA | 254 | 549 | 1144 | 0.70 | 1.51 | 3.14 |
| GOF | M-MS | O | 10897 | 10184 | 9410 | 3.14 | 2.94 | 2.71 |
| GOF | M-MS | H | 1684 | 2874 | 1256 | 6.33 | 10.80 | 4.72 |





| | | | | | | | | |
|------|------|----|-------|-------|-------|--------|--------|--------|
| GOF | M-MS | A | 736 | 259 | 2651 | 6.98 | 2.46 | 25.16 |
| NBP | S | O | 109 | 107 | 107 | 0.03 | 0.03 | 0.03 |
| NBP | C | O | 171 | 171 | 156 | 0.00 | 0.00 | 0.00 |
| NBP | M | O | 10707 | 7587 | 8262 | -8.97 | -6.35 | -6.92 |
| NBP | M | HA | 1866 | 4985 | 4310 | 5.12 | 13.68 | 11.83 |
| NBP | M-MS | O | 8581 | 4459 | 5727 | 0.62 | 0.32 | 0.41 |
| NBP | M-MS | H | 2995 | 4353 | 1396 | 44.65 | 64.90 | 20.81 |
| NBP | M-MS | A | 4940 | 7704 | 9393 | 63.01 | 98.28 | 119.81 |
| WGB | S | O | 369 | 368 | 368 | 0.11 | 0.11 | 0.11 |
| WGB | C | O | 387 | 380 | 383 | 0.01 | 0.01 | 0.01 |
| WGB | M | O | 11496 | 10282 | 10142 | 0.51 | 0.46 | 0.45 |
| WGB | M | HA | 5782 | 6996 | 7136 | 15.86 | 19.19 | 19.58 |
| WGB | M-MS | O | 2178 | 2283 | 1833 | 0.45 | 0.47 | 0.38 |
| WGB | M-MS | H | 2144 | 948 | 1127 | 21.60 | 9.55 | 11.35 |
| WGB | M-MS | A | 6545 | 7636 | 7906 | 62.95 | 73.44 | 76.04 |

**Author contribution**

AH, NE, AD, MK, WL, AT, SV, SS, CS and PH conducted the field work. AH and NE assembled the data. AH evaluated the flux data, produced the figures and wrote the original draft of the manuscript. NE, HB and AP conducted the upscaling
calculations. SS, CS and POJH obtained funding for the data collection. All authors contributed to the writing of the final version of the manuscript.

**Competing interests**

SV is a member of the editorial board of the journal. The authors declare that they have no other conflicts of interest.

**Acknowledgements**

We thank the captains and crews on the research vessels Skagerak, Fyrbyggaren, KBV005, Aranda, Poseidon and Alkor for skilful support at sea, .Lena Viktorsson and Martin Hansson at the Swedish Meteorological and Hydrological Institute (SMHI) for providing data on the extend of oxygen-depleted waters. This study was financially supported by the Swedish Research Council (grants number 621-2007-4641, 621-2012-3965, and 2015-03717 to POJH) and by the Swedish Agency for Marine





and Water Management (grants number 3159-19, 2535-20, and 1484-2022 to POJH). AH was funded by a junior postdoctoral
fellowship from FWO (project nr 1241724N). The Swedish Agency for Marine and Water Management, who have funded the
study, have been granted a permit to disseminate the data by the Swedish Maritime Administration.

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
