# Peer review of "In situ-measured benthic fluxes of dissolved inorganic phosphorus in the Baltic Sea"

_Earth System Science Data, 2025_

## Author Comment (AC1)

**Our answers are provided in blue.**

I find the manuscript highly informative, as it provides a comprehensive long-term dataset on benthic phosphorus fluxes in the Baltic Sea, offering valuable insights into nutrient cycling and contributing to more effective management of coastal eutrophication. I also appreciate the discussion of the study's limitations and the well-considered suggestions for future research directions.

We thank the reviewer for their feedback, which has helped us to clarify several methodological points.

**General Comments:**

1. In the first sentence of the abstract, I recommend avoiding exclusive emphasis on coastal eutrophication, as phosphorus recycling is a process of broader significance across marine ecosystems. Furthermore, the majority of the data presented in the study are not from coastal environments, so a more general framing would be more appropriate.

   We agree, and have removed the word "coastal".

2. Please provide the source of the oxygen data used in Figure 1. If the data were collected as part of this study and extrapolated, a detailed description of the methodology should be included in the Methods section.

   We thank the reviewer for pointing out this omission. As described in the main text, the extent of hypoxic and anoxic conditions was taken from the SMHI oxygen survey from 2021 (Hansson and Viktorsson, 2021) – this information has been added to the figure caption.

3. According to many standard recommendations for phosphate sample collection and preservation, filtration (typically at 0.45 µm) is required to remove suspended particles and microorganisms. If unfiltered samples were used, a turbidity blank must be applied to correct for potential interference. Additionally, acidification is generally not advised for phosphate preservation. Given that various collection and preservation methods appear to have been used in this study, please discuss the uncertainties this introduces into the fluxes and extrapolations.

   All samples were filtered before analysis; the text has been corrected. Samples are routinely acidified to avoid the potential precipitation of iron oxides, which could scavenge phosphate (Bray et al., 1973, doi: 10.1126/science.180.4093.1362; Slomp et al., 1996, doi: 10.1357/0022240963213745).

The fluxes are not affected by potential differences in concentrations between measurements between methods, as they are calculated based on concentration changes. We have clarified this in the text.

L177-179: "All methods used to measure the DIP are routinely used for seawater, so no differences in concentrations are expected between methodologies. Furthermore, the resulting fluxes are not affected by methodological differences, as they are calculated from the change in DIP concentration in samples measured with the same methodology."

4.  I assume that samples were analyzed in different laboratories. If so, please provide detailed information on the accuracy and precision of the analytical methods used, as this can significantly affect the comparability of the results and calculations.

    While the samples were analysed in different laboratories, we have clarified in the text that the same method was used and similar analytical precisions were obtained:

    L175-177: "Concentrations of DIP were, in all cases, determined by the ammonium molybdate method using segmented flow colorimetric analysis (Koroleff, 1983) with an analytical precision better than 3%."

5.  The authors state that the dataset has limited use for interpreting seasonal trends. However, it would be interesting to deliberate a bit on the temporal variability at stations where monthly coverage was substantial, such as KH104.

    We have added a figure to the appendix showing the DIP fluxes as a function of month at sites that were sampled in at least three different months (Figure A3).

    Station KH104 is the only site in the dataset that has been sampled in more than two seasons. The DIP flux patterns at this station have been discussed extensively by Ekeroth et al. (2016, doi: 10.1016/j.jmarsys.2015.10.005), and changes to the $O_2$ conditions in the bottom water were concluded to have a larger impact than seasonality. We have clarified in the text why seasonal patterns in the data are not discussed.

    L248-250: "At 14 stations, sampling occurred during three or four different months (Fig. A3). However, except for at station KH104, samplings were conducted across different years, only during two seasons, and aimed at observing shifts in $O_2$ conditions. As a result, seasonal trends cannot be determined from this dataset."

---

## Author Comment (AC2)

**Our answers are provided in blue.**

The manuscript presents a dataset on benthic dissolved inorganic phosphorus (DIP) fluxes in the Baltic Sea, highlighting spatial and temporal variations in relation to the extent of deep-water deoxygenation and the underlying sediment types. The dataset is remarkable, particularly because it was obtained through in situ measurements. I especially appreciated the effort made to harmonize the data and to ensure its robustness. This manuscript represents a valuable contribution to the marine biogeochemistry community as well as to researchers interested in coastal deoxygenation processes.

That said, several points would benefit from clarification, particularly to ensure that readers who are not specialists of the Baltic Sea can easily follow and fully grasp the information provided.

We thank the reviewer for their comments, which have enhanced the manuscript's accessibility for a wider readership.

**Comments**

L38-40: The sentence is ambiguous. It's not clear whether you mean that the basins differ from each other, or that they are similar to each other but differ collectively from other semi-enclosed seas. Could you please clarify?

They differ from each other; we have clarified this in the text.

L39-41: "Importantly, the basins of the Baltic Sea differ substantially from each other in environmental conditions as well as in input rates and sources of organic matter, macro and micronutrients…"

L43: The addition of surface and subsurface currents would be interesting on this map. It would help visualize circulation. Colored arrows according to salinity could make it easier to understand. L47: "The varied environmental conditions in the Baltic Sea are largely related to strong salinity gradients" - Please add the information in the map.

We have added arrows indicating currents to Figure 1.

L45: Please define this acronym. Not everyone is familiar with this agency. Please consider that some readers are not from this part of Europe.

We thank the reviewer for pointing out this oversight, we have added the full name of HELCOM to the caption.

L45: Why did you specifically choose these 3 years? Indicate data source

The caption has been expanded and now states that the three years were chosen to represent small, medium and large extents of hypoxia and anoxia, and the data source has been added.

L60: "small coastal basin" - Please clarify and give examples that can be seen on the map. If it's really more "coastal", please indicate so.

We have modified the sentence to specify that the highest sediment accumulation rates are found in archipelago areas.

L63-65: "However, local conditions strongly affect sedimentation and the highest sediment accumulation rates are found in small coastal basins, particularly in archipelagos (Mitchell et al., 2021)."

The precise patterns in sediment accumulation rates are discussed extensively in the cited study (Mitchell et al., 2021), including detailed maps. Further details about sediment accumulation rates are outside the scope of this study.

L64: "The catchment area consists primarily of forests, and the riverine input of terrestrial material is relatively large " - Quantify. Add an average load or average flow for comparison with other environments.

The introduction describes the relative differences between basins. Data coverage and methods for estimating the load of riverine particulate and dissolved organic matter differ considerably over the Baltic Sea are, which makes quantifications uncertain. We have therefore opted to leave the text as it is.

L108: "the big and small University of Gothenburg landers" - Can you add size? This is of interest to those interested in the effect of size on the value and accuracy of flows. add the Table 2 here.

We have restructured the text and now start the material and methods section with a subsection about the benthic chamber lander systems (Section 2.1), which includes Table 2 (now Table 1). The sizes of the chambers are stated in the table. As suggested by Reviewer 2 and Reviewer 3, we also mention that the chamber design does not impact the flux.

L118-119: "Previous intercalibration studies have shown that the chamber design does not significantly impact the measured flux (Tengberg et al., 2004, 2005)."

L113: Do the colors of the dots also refer to the type of sediment? Are the differences related to scale?

The reviewer is correct; the colour of the markers shows the sediment type as determined at sampling. We have added this to the caption.

L124-125: "... field observations of sediment cores occasionally indicated that the actual sediment type differed from that inferred from EMODnet." - Answer to my comment above. I'll add the info anyway in the caption.

Added.

L130: Surface? Surficial.

We have rewritten the sentence.

L 154: "Indeed, the median OC content at the sediment surface (top 0-2 cm) ... "

L147: see a previous comment

Changed according to the suggestion.

L152-153: I'm not sure I understand what was done to ensure the success of incubation. Not just based on oxygen consumption? Please clarify by adding more information.

We have added further information about how the success of the incubations was confirmed.

L114-118: "Besides being used to calculate $O_2$ fluxes, the $O_2$ data also confirmed that no leakages or disturbances occurred during the incubations. Depending on the lander system, successful incubations were further corroborated by, e.g., measurements of turbidity and pressure within the chambers to confirm that resuspension did not occur and that syringes were triggered at the right time, or retrieval of the incubated sediment at the end of the deployment for visual observation (Kononets et al., 2021; Sommer et al., 2006; Witbaard et al., 2000). "

L155-157: Samples only dedicated to DIP or other nutrient ? For DIP too? or for trace metal or Fe?

The text describes how samples analysed for DIP were treated. Other solutes were measured simultaneously, from the same or different aliquots. We have clarified this in the text.

L179-180: "Other solutes were measured in parallel to DIP; these data are presented in the studies listed in Table 2."

L195: must be presented beforehand.

The full name of HELCOM is now written out at its first occurrence in the text.

L199: ??

EMODnet, the source of the sediment type map, is presented in section 2.2.

L210: figures are not called up in order.

We have changed the text and now refer to Section 3.1 instead of Figure 6.

L217-218: impressive!!

Thank you!

L244: I guess it doesn't matter what year it is. You'd have to add a sentence to explain that it's compile, regardless of the year of collection... it refers to the station and the oxygen concentration.

We have changed the caption.

L272: "Figure 4 Compilation of all individual fluxes of dissolved inorganic phosphorus (DIP) with increasing depth..."

L249: whatever the O2 concentration

We have changed the caption.

L278: "Figure 5 Sediment-water fluxes of DIP in the Baltic Sea, as averages of all oxygen conditions and years per station."

Table A1: 1% is missing

This was due to a rounding error and has been corrected.